# The Gut–Liver Axis in Chronic Liver Disease: A Macrophage Perspective

**DOI:** 10.3390/cells10112959

**Published:** 2021-10-30

**Authors:** Kevin De Muynck, Bart Vanderborght, Hans Van Vlierberghe, Lindsey Devisscher

**Affiliations:** 1Gut-Liver Immunopharmacology Unit, Department of Basic and Applied Medical Sciences, Liver Research Center Ghent, Ghent University, 9000 Ghent, Belgium; k.demuynck@ugent.be (K.D.M.); bart.vanderborght@ugent.be (B.V.); 2Hepatology Research Unit, Department of Internal Medicine and Pediatrics, Liver Research Center Ghent, Ghent University, 9000 Ghent, Belgium; hans.vanvlierberghe@uzgent.be

**Keywords:** macrophage, chronic liver disease, gut-liver axis

## Abstract

Chronic liver disease (CLD) is a growing health concern which accounts for two million deaths per year. Obesity, alcohol overconsumption, and progressive cholestasis are commonly characterized by persistent low-grade inflammation and advancing fibrosis, which form the basis for development of end-stage liver disease complications, including hepatocellular carcinoma. CLD pathophysiology extends to the intestinal tract and is characterized by intestinal dysbiosis, bile acid dysregulation, and gut barrier disruption. In addition, macrophages are key players in CLD progression and intestinal barrier breakdown. Emerging studies are unveiling macrophage heterogeneity and driving factors of their plasticity in health and disease. To date, in-depth investigation of how gut–liver axis disruption impacts the hepatic and intestinal macrophage pool in CLD pathogenesis is scarce. In this review, we give an overview of the role of intestinal and hepatic macrophages in homeostasis and gut–liver axis disruption in progressive stages of CLD.

## 1. Introduction

Chronic liver disease (CLD) poses a major global health problem with a mortality rate of approximately two million deaths per year worldwide. The term “CLD” encompasses a group of disorders that are hallmarked by persistent liver inflammation and progressive fibrosis which can progress to hepatocellular carcinoma (HCC) [1,2]. Over the years, increasing attention has been paid to the role of the gut in CLD. Liver cirrhosis is characterized by intestinal barrier disruption and a disturbed gut–liver crosstalk has been implicated in non-alcoholic fatty liver disease (NAFLD), alcoholic liver disease (ALD), primary sclerosing cholangitis (PSC), liver fibrosis/cirrhosis, and HCC [3,4,5,6,7]. While a leaky gut, bacterial dysbiosis, and bile acid (BA) alterations are considered as obvious indications of gut–liver axis disruption, the role of the immune system remains underexplored [8,9].

The emergence of macrophages (MFs) as cell-based therapy for liver cirrhosis emphasizes the importance of these innate immune cells in CLD [10]. MFs are key players in intestinal and hepatic inflammation by promoting initial inflammatory responses as well as facilitating recovery from disease. Multiple preclinical data strengthen the potential of specific MF populations as therapeutic target and indicate that MFs can serve as therapy for intestinal inflammation and liver disease [11,12,13]. Peripheral infusion of autologous monocyte-derived MFs (MoMFs) was reported to be safe and feasible in cirrhotic patients and may form the basis for new targeted cell-based therapy [10]. In this context, an in-depth understanding of the role of MFs in liver diseases, and specifically in gut–liver communication in the context of CLD, is important and may lead to new important insights and more fine-tuned pharmacological approaches.

This review focusses on the composition of the intestinal and hepatic MF pool and their subset-specific functions in homeostasis and gut-liver axis disruption, with specific elaboration on NAFLD, ALD, PSC, fibrosis/cirrhosis, and HCC.

## 2. The Gut–Liver Axis

The gut–liver axis refers to the close anatomical and physiological relationship between the gut and the liver. The liver produces BAs which are secreted into the duodenum, where they fulfill an essential role in lipid metabolism and microbial homeostasis, and subsequently transported back to the liver after ileal absorption via the portal vein (the enterohepatic circulation) [8,14,15]. As such, the liver and the gut are continuously connected to each other by the systemic circulation and the exterior, and are therefore strategically organized as first and second line immunoregulatory surveillance system (Figure 1) [16,17].

The gut mucosal firewall comprises several physical and immunological barriers. On the luminal side, the outer mucus layer consists of a vast array of epithelial cell-derived antimicrobial peptides, along with B cell-secreted IgA, which both exert modulating effects on the commensal gut microbiota. The inner mucus layer resides on top of an epithelial monolayer of enterocytes, which prevents bacterial translocation (BT) through the expression of paracellular transport-regulating tight junction (TJ) proteins [8,17,18]. In addition, this epithelial barrier also contains a heterogeneous population of intraepithelial lymphocytes which contribute to the luminal immune surveillance [19]. The lamina propria (LP), located directly below the intestinal epithelium, contains both diffusely distributed and Peyer’s patch (PP)-associated MFs, dendritic cells (DCs), and T and B lymphocytes, together organized as the gut-associated lymphoid tissue (GALT). Translocated bacteria and their microbial-associated molecular patterns (MAMPs) are either eliminated by LP-associated MFs or captured by DCs, resulting in quiescent toll-like receptor (TLR)/nucleotide-binding oligomerization domain-like receptor (NLR) signaling. Antigen-loaded DCs present processed MAMPs to naive T and B cells in the PPs or mesenteric lymph nodes. After priming, the antigen-specific effector T and B cells migrate to the LP, where T cells directly restrict the passage of specific microbes and B cells secrete antigen-specific IgA into the outer mucus layer. Importantly, all these immune-mediated responses are controlled and restricted to the intestinal mucosa and the draining mesenteric lymph nodes to maintain immune tolerance towards harmless dietary and commensal bacterial antigens [20,21,22].

The liver functions as a second firewall, with a slow-flowing vascular system harboring a pool of specialized intra- and perisinusoidal liver-resident cells capable of antigen presentation without translocation towards lymph nodes. In addition to liver sinusoidal endothelial cells, Kupffer cells (KCs) and hepatic DCs, which are all considered as conventional liver antigen-presenting cells, hepatocytes, and hepatic stellate cells (HSCs) also contribute to hepatic immune tolerance as non-conventional antigen-presenting cells. KCs (resident hepatic MFs) and hepatic DCs are strategically located in the sinusoids to clear the portal blood from gut-derived microorganisms and MAMPs. Liver sinusoidal endothelial cells form a fenestrated barrier, which facilitates interaction between the sinusoidal blood and perisinusoidal hepatocytes and HSCs (also known as Ito cells) in the space of Disse. In addition, this intra/perisinusoidal niche contains a broad repertoire of lymphoid cells, including T cells, B cells, NK cells, and NKT cells. The slow pace of the sinusoidal blood flow effectuates continuous interaction between lymphocyte populations and hepatic antigen-presenting cells, however without eliciting an immune response. Liver immune tolerance is characterized by an anti-inflammatory cytokine milieu, including high levels of IL-10 and TGF-β, downregulation of co-stimulatory molecules and upregulated expression of co-inhibitory molecules, including programmed death ligand 1 (PD-L1) [16,23,24].

The liver impacts the intestinal environment through secretion of BAs, which contributes to intestinal eubiosis by preventing pathogenic microbial overgrowth [8,14,15,25]. In addition, BAs have differential roles in macronutrient metabolism, intestinal barrier function, as well as anti-inflammatory immune responses [8,14]. In the gut, microbiota metabolize primary BAs into secondary BAs which undergo enterohepatic cycling via uptake in the terminal ileum [8,14,25,26]. BA homeostasis is tightly regulated via signaling pathways involving nuclear receptors, including farnesoid X receptor, pregnane X receptor, and vitamin D receptor and Takeda G protein-coupled receptor 5, which are expressed in both intestinal and hepatic tissue [8,14,27].

## 3. Macrophages during Gut–Liver Axis Homeostasis

MFs (‘big eaters’ in Greek) are named after their prototypical function of phagocytosis and play a pivotal role in inflammation and infection [28]. The emergence of high-resolution techniques such as multicolor flow cytometry and scRNA-seq, in conjunction with fate mapping, lineage tracing, and cell-specific depletion models, has drastically changed our understanding of MF biology in the recent years [29,30,31,32]. A substantial amount of research now provides evidence that MFs exert crucial roles in tissue development, homeostasis, and repair [28]. Importantly, MFs are a heterogeneous population with respect to ontogeny, renewal capacity, plasticity, polarization, and (niche-specific) functionality, and should hence forth consistently be considered in their tissue-specific context [33,34]. The MF pool composition is thus largely dependent on the microenvironment it resides in, and can consequently vary greatly between different tissues, and between homeostatic and pathological conditions in a particular tissue [35]. These paradigm shifts have triggered a call to abandon the classical M1/M2 dichotomy [36] in favor of a more specific function-based nomenclature in order to classify MFs based on the expanding spectrum of context-dependent polarization states [37,38,39].

MFs either originate from the embryonic yolk sac, the fetal liver, or differentiate from circulating blood monocytes. Intestinal MFs were long thought to be solely replenished by bone marrow-derived monocytes. However, recent studies revealed that the intestinal tract comprises a heterogeneous population of MFs with distinct location-specific phenotypes and functions and highlighted embryonic-derived micro-anatomical subsets with self-renewal capacity (Table 1). The LP contains the largest intestinal MF pool, which is most abundant in the colon given the higher microbial load [40,41,42,43]. In the LP, MFs are continuously replenished via the ‘monocyte–macrophage waterfall’, a multistep process in which extravasated Ly6C^hi^ CX3CR1^int^ MHCII^−^ monocytes give rise to mature Ly6C^lo^ CX3CR1^hi^ MHCII^+^ MFs [40]. Steady-state LP MFs are highly phagocytic and remain quiescent in response to food antigens or harmless microbes. They participate in oral tolerance induction by sampling the gut lumen and presenting antigens to DCs. Self-renewing subsets were found to be located near intestinal neurons and blood vessels. Below the LP, neuron-associated MFs in both the submucosal and myenteric plexus were reported to highly express genes that are enriched in microglia and are proposed to be crucial in neuronal cell population maintenance. Submucosal self-renewing blood vessel-associated MFs were shown to upregulate angiogenesis-associated genes and are suggested to contribute to sustaining the integrity of the gut vascular barrier [44]. However, a recent study identified blood-vessel associated MFs specifically located in the villi that were derived from circulating monocytes [45]. Moreover, a specific CD169^+^ subset was also closely associated with the vasculature, but less is known regarding ontogeny and transcriptional profile of these MF subsets [46]. Other niche-specific MFs located near crypts [47] and PPs [48] have also been described in mice.

In humans, analogues of mucosal monocyte-MF populations have been described very recently. scRNA-seq profiling revealed 11 monocyte-MF clusters which showed gene expression profiles comparable with their murine analogues. Pseudotime trajectory analysis showed a maturation pathway reminiscent of the ‘monocyte–macrophage waterfall’ described in mice [49]. In line, a differentiation trajectory in human intestinal mucosa, where CD14^hi^ CCR2^+^ CD11c^hi^ monocytes mature into CD14^lo^ CCR2^−^ CD11c^lo^ MFs, was described earlier [50,51]. In contrast to mice, however, MF differentiation in the human gut mucosa was described to be characterized by downregulated CX3CR1 expression [51]. The microniche-specific spatial context of these human mucosal analogues still needs to be investigated, and human counterparts of intestinal neuron-associated MFs are yet to be identified.

**Table 1 cells-10-02959-t001:** Intestinal macrophage subsets in mouse and human.

Macrophage Subset (Niche/Development)	Specific Markers	Intestinal Segment	Intestinal Wall Substructure	Reported Function	Properties	Reference(s)
**MOUSE**						
Epithelium-associated	CD11c, CD121b	Colon	Lamina propria	Unknown	Regulated by microbiota	[52]
Blood vessel-associated	ADAMDEC1, CD169	Small intestine, Colon	Lamina propria	Blood vessel maintenance	Self-renewing	[44,45,46]
Neuron-associated	Fcrls	Small intestine	Muscularis layers	Facilitate enteric neuron survival	Self-renewing	[44]
Crypt-associated	(CD169)	Small intestine, Colon	Lamina propria	Unknown	Self-renewing	[44,46]
Peyer’s patch-associated	CD4, TIM4	Small intestine	Lamina propria	TIM4^+^: phagocytic TIM4^−^: microbial defense	TIM4 expression can discriminate location	[48]
Unknown	CD4, TIM4	Colon	Unknown	Unknown	Long-lived	[43]
**HUMAN**						
Late intermediate (Clusters M4–6)	CD11c^int^ C1QC^int^ CD206^+^ CD11a^+^ CD1c^int^	Small intestine, Colon	Lamina propria, excl. ILF and GALT	M5: ↑ NF-κB, TNF-α, EGFR, and MAPK signaling	M4,6: ↑ JAK-STAT, Wnt signaling; (less inflammatory than M5, 7) M5: inflammatory + ↑ EGFR and MAPK signaling	[49]
Mature (Clusters M7–8)	CD209^hi^, C1QC^int^, CD206^+^ M7: CD14^hi^, LYVE1^+^ CD169^+^ FOLR2^+^ MAF^+^	Small intestine, Colon	Lamina propria, excl. ILF and GALT	M7: ↑ NF-κB, TNF-α, TGF-β, EGFR, and MAPK signaling M8: ↑ JAK-STAT and Wnt signaling	M7: inflammatory + similar to BVAMs, cfr. Honda et al., 2020	[49]
Mature-like (Clusters M9–10)	MHCII^hi^ C1QC^hi^ CD209^lo^ CD163^lo^ CD206^−^ M9: CD11c^+^ CD4^+^ C2^+^ ADAMDEC1^+^ M10: CD11c^−^ FTL^+^ RBP1^+^ PLP1^+^ PMEPA1^+^ GPM6B^+^ SPARC^+^ ADIRF^+^ SCD^+^	Small intestine, Colon	Lamina propria, excl. ILF and GALT	M9: ↑ JAK-STAT, Wnt, TGF-β signaling M10: genes in iron metabolism, fibrosis, adipocyte-regulation; not involved in efferocytosis	M9: cfr. De Schepper et al., 2018 and Shaw et al., 2018	[49]

Recent studies have shown that both murine and human hepatic MFs also comprise different populations with context-dependent phenotypes and functions (Table 2) [53,54]. In homeostasis, liver resident MFs or KCs are the most highly represented immune cells of the liver, comprising up to 80% of the body’s total MF pool [39,55]. Located in the sinusoids, they are in continuous interaction with other non-parenchymal hepatic cells, including other intrasinusoidal myeloid and lymphoid immune cells, and orchestrate immunological tolerance through phagocytosis of both systemic and gut-derived pathogens, high expression of pattern recognition receptors and production of several immunoregulatory mediators, including IL-10 and PD-L1. However, Blériot et al. recently described a minor CD206^hi^ESAM^+^ KC subset that is predominantly wired with metabolic functions and expresses surface markers previously thought to be exclusive for liver sinusoidal endothelial cells. These KCs present with a different transcriptional profile compared to their immunomodulatory counterparts, but express the KC core signature and are embryonic derived [56]. Murine KCs primarily originate from fetal yolk-sac-derived progenitors and self-renew via M-CSF-dependent proliferation [37]. However, experimental KC depletion studies have shown that monocytes are able to adopt a KC-resembling phenotype with the capacity to self-renew when the KC niche becomes available, giving rise to monocyte-derived KCs (MoKCs) [57,58]. In addition to the presence of Clec4F^+^TIM4^+^ KCs, small amounts of Clec4F^−^TIM4^−^ MoMFs are also present in healthy conditions [59,60]. Lastly, other distinct MF subsets, including liver capsular MFs and peritoneal MFs, are also considered part of the hepatic MF pool [61,62]. The characterization of these various hepatic MF subsets, especially in terms of their surface marker expression, has thus far most extensively been performed in mice [39,55]. In human livers, scRNA-seq only recently enabled the identification of CD68^+^CD163^+^MARCO^+^TIMD4^+^ KCs (which share a highly conserved transcriptional signature with their murine counterparts), CD68^+^MARCO^−^ MoMFs and CD14^+^ infiltrating monocytes [53,63,64,65].

## 4. Macrophages during Gut-Liver Axis Disruption in Chronic Liver Disease

CLD is associated with a shift in hepatic MF populations [59,60,125,126,127,128]. When liver MFs are exposed to high concentrations of damage-associated molecular patterns (DAMPs)/MAMPs, they polarize towards a pro-inflammatory phenotype, resulting in subsequent attraction of other inflammatory cells through the production of pro-inflammatory cytokines and chemokines (Figure 1) [37,39,129]. KC activation was generally assumed to drive the initiation of inflammation. However, an increasing number of studies are showing that the contribution of KCs and MoMFs to disease initiation and progression is highly etiology- and model-dependent [77,79]. An important MAMP associated with CLD is bacterial lipopolysaccharide (LPS), which activates hepatic MFs through interaction with TLR4 [130,131]. Indeed, in NAFLD patients, LPS-induced activation of liver MFs is associated with inflammation and fibrosis, and both TLR4 knockout (KO) and clodronate-mediated hepatic MF depletion attenuate experimental NASH [132,133]. Preventing hepatic MF activation by targeting the LPS–TLR4 axis may thus represent a valid therapeutic strategy for CLD, especially since pharmacological inhibition of TLR4 has already been shown to ameliorate liver injury and inflammation in experimental fibrosis/cirrhosis [130,134]. In addition to LPS, other gut-derived bacterial metabolites are involved in hepatic MF-mediated immunity as well [130]. The intestinal microbiota-dependent tryptophan metabolites tryptamine and indole-3-acetate reduce the production of proinflammatory cytokines by liver MFs and, importantly, are depleted in mice receiving a high fat diet (HFD) [135]. Intestinal dysbiosis can also be of fungal nature. Chronic ethanol administration in mice leads to an altered intestinal mycobiome, with increased translocation of β-glucan to the liver, leading to hepatic MF-induced liver injury and inflammation [136]. An effective strategy to prevent MAMP-mediated MF activation in CLD might be restoration of eubiosis using broad-spectrum antibiotics, probiotics, and fecal microbiota transfer [11,134]. In mice receiving HFD, modulation of the gut microbiome through the administration of inulin was found to reduce TLR4-mediated hepatic MF activation and prevent NAFLD development [137,138].

Interestingly, in mice, liver disease is associated with a depletion of KCs and enrichment of liver MoMFs [59,60]. These MoMFs are derived from infiltrated bone marrow-derived CCR2^+^ CX3CR1^lo^ Ly6C^hi^ monocytes, whose recruitment is dependent on the CCL2–CCR2 axis [37,39,55,139]. Infiltrated monocytes can contribute to both the development and resolution of hepatic inflammation [39,129]. Following their extravasation in the liver, monocytes develop into pro-inflammatory Ly6C^hi^ MoMFs, while upon resolution, these pro-inflammatory MoMFs can differentiate into mature CCR2^−^ CX3CR1^hi^ Ly6C^lo^ MoMFs, with a more restorative phenotype, stressing the relevance of timely analysis in a given experimental model [39,129,140]. Since the human liver appears to harbor a similar subdivision of hepatic MF subpopulations in both health and liver disease [141], it will be of great interest to assess the extent to which context-dependent MF subsets and phenotype-specific functions are preserved across species.

Intestinal MFs have been extensively investigated in experimental and human inflammatory bowel disease (IBD) [13,40,51,142,143] (reviewed in [13]). Chronic inflammatory intestinal conditions are associated with a massive influx of monocytes into the LP, but MFs have equally been shown to be instrumental during resolution of inflammation [143] and intestinal fibrosis [144]. CLD is frequently associated with disruption of intestinal homeostasis and inflammation, and gut MFs have been proposed to express factors capable of enhancing intestinal permeability in cirrhosis patients [145]. However, to what extent intestinal monocyte-MF populations are altered during various stages of CLD progression, especially in context of a dysregulated gut–liver axis, is far less known.

In the following sections, we elaborate on the dysregulated gut–liver axis in selected CLDs with varying etiologies. We discuss gut microbiota and gut barrier disruption and focus on intestinal and hepatic MF populations in human and experimental CLD. Additionally, we briefly highlight how the hepatic MF pool can be influenced from a therapeutic point of view.

### 4.1. Non-Alcoholic Fatty Liver Disease

NAFLD is predicted to become the leading cause of end-stage liver disease and hence, in addition to ALD, the most common indication for liver transplantation around the globe [146,147,148]. The current prevalence is nearly 25% worldwide and although typically associated with Western countries, the highest rates are reported in South-America and the Middle-East [147]. NAFLD encompasses a disease spectrum ranging from simple steatosis to steatohepatitis (NASH) and end-stage liver disease. Up to 20% of people with NAFLD develop NASH, and this evolves to cirrhosis in 10–20%. Around 2–12% of NASH-cirrhosis patients develop HCC [149,150,151]. NAFLD is strongly associated with a sedentary lifestyle, Western diet, type 2 diabetes, cardiovascular disease, and obesity. However, NAFLD is also observed in 10–20% of lean Europeans/Americans [146,152]. The amount of visceral adipose tissue is a risk factor for NAFLD/NASH development [153] and adipose tissue MFs have shown to be instrumental in the NAFLD disease spectrum. Discussion of adipose tissue MFs is beyond the scope of this manuscript, but they are extensively reviewed elsewhere [154,155,156,157,158]. MF activation in general is associated with disease severity in clinical NASH [159,160,161] and, moreover, correlates with dietary patterns [162].

Dietary habits pose a major risk factor for NAFLD development and can have a significant impact on the gut–liver axis. Western diet consumption for one month induces endotoxemia [163] and is linked to intestinal dysbiosis, both in human and experimental NAFLD [164,165]. Intestinal dysbiosis is observed in various NAFLD models and a growing number of studies are describing specific microbial signatures in NAFLD patients at various stages of disease progression [166]. Recently, Yuan et al. were able to pinpoint a high alcohol-producing *Klebsiella pneumonia* strain as a NAFLD-driving factor in mice, providing mechanistic insights on how dysbiosis can contribute to a fatty liver [167,168]. However, in human NAFLD, clearly defined causal roles for specific gut microbial organisms remain elusive.

BAs mediate fat metabolism and HFD induces changes in the BA pool, which in turn can contribute to intestinal dysbiosis [169,170]. Apart from increased fat consumption, the intake of sugar-sweetened beverages rich in fructose also strongly correlates with NAFLD as shown in a recent meta-analysis by Farzaneh et al. [171]. Fructose was reported to cause protein nitration of intestinal TJs, resulting in increased gut leakiness, endotoxemia, and steatohepatitis [172]. In addition, fructose was shown to directly modulate the intestinal microbiota resulting in dysbiosis and further enhanced gut leakiness and increased endotoxemia [173]. Barrier disruption and dysbiosis increase endotoxin exposure and bacterial translocation into the LP, and increase the microbial load of intestinal MFs [174]. Mice fed a HFD showed decreased numbers of intestinal CX3CR1^+^ MFs, which increased susceptibility to HFD-induced liver injury due to impaired barrier function [175]. In addition, HFD was reported to aggravate colitis [176]. Inversely, colitis and associated barrier disruption aggravated HFD-induced experimental NAFLD, providing evidence that gut-derived hits can drive the progression from NAFLD to NASH [177,178]. Future studies should focus on characterizing the effects of HFD on the myeloid cell pool in the healthy and inflamed colon, both in experimental models and humans, to unravel how changes in intestinal MF subsets might extend to hepatic NAFLD pathophysiology.

CD68^+^ MFs accumulate in periportal areas of human NAFLD patients with simple steatosis and further increase with NASH severity [118,179,180]. Very recently, hepatic macrophage scavenger receptor 1 (MSR1 or CD204) expression was shown to correlate with steatohepatitis, while serum levels of MF activation markers, including sCD163 and sSiglec-7, mostly correlated with NASH-associated fibrosis [159,181]. In experimental NAFLD, early clodronate- or gadolinium chloride-mediated MF depletion prevented disease progression indicating the role of hepatic MFs in NAFLD progression [182,183,184]. Compositional alterations in the hepatic MF pool during NAFLD are commonly observed, but conclusions regarding the dynamics and properties of specific MF subsets often do not align, in part explained by non-uniform identification of specific hepatic MF subsets [59,86,98,184,185,186,187,188]. 

Currently, resident KCs are characterized by simultaneous expression of Clec4F and TIM4. In NAFLD/NASH models, numbers of resident KCs were reported to be decreased [94,95,97,98]. Furthermore, this demise of resident KCs was suggested to drive MoKC recruitment, at least in the methionine/choline-deficient diet (MCD) model [97]. In another study using the MCD model, Devisscher et al. demonstrated that most of these infiltrated MoKCs were short-lived and only a fraction gained TIM4 expression during recovery from MCD [59]. In line, Tran et al. showed that only 20% of MoKCs acquire TIM4 expression, but in addition they reported that MoKCs gain self-renewing properties and remain present in the liver even after 12 weeks recovery from MCD [97]. To date, there is controversy regarding the function of KCs and MoKCs. KCs are conventionally considered the initial inflammatory responders, activated by increased levels of gut-derived MAMPs and DAMPs released from (toxic) lipid-overloaded dying hepatocytes [184]. This triggering, initiating role of KCs was recently contested by Remmerie et al. who showed that resident KCs were not pro-inflammatory in a Western diet NAFLD model. However, the authors did not investigate early KC activation, which might yield different results [94,189]. In an acute liver injury model, KCs were also shown not to be the inflammatory macrophage population, further refuting an important proinflammatory role for KCs in liver injury initiation and progression. In line, Tran et al. reported that MoKCs were more inflammatory compared to resident KCs, and that resident KCs were more inflammatory in the MCD vs the HFD model [97]. MoKCs displayed a reduced capacity for lipid-handling compared to resident KCs [94,97]. Of note, Remmerie et al. suggested that Clec1b (=human CLEC2), a marker used by Tran et al. to define KCs, can serve as a preMoKC marker, as its expression precedes Clec4F [94]. The functional heterogeneity of the KC pool during NAFLD was very recently reported to be even more complex than previously reported. Blériot et al. described a metabolically wired KC subset that prevents HFD-induced weight gain and oxidative stress when conditionally depleted [56]. Future kinetic and conditional depletion studies of each of these described KC subsets in a variety of NAFLD models are needed to describe their dynamics and more clearly define the roles of KC subsets and (pre)MoKCs during NAFLD initiation and progression.

In addition to the generation of MoKCs, monocytes can also give rise to MoMFs, which is most pronounced upon liver injury [93,190]. MoMFs derive from circulating monocytes that are attracted to the liver via the CCL2–CCR2 axis. Therapeutic inhibition of the CCL2–CCR2 axis ameliorates experimental NASH [86,191,192]. In patients in a phase 2b study, cenicriviroc (CVC), a dual CCR2/CCR5 inhibitor, was reported to be most efficacious in patients with more advanced degrees of NASH-associated fibrosis (CENTAUR study; NCT02217475). However, the CVC phase 3 study (AURORA; NCT03028740) [193] was recently terminated due to lack of efficacy. Thus, inhibiting monocyte recruitment does not seem to be an effective treatment strategy for NASH, based on the data available so far.

Several studies have independently identified CD9^+^ CD63^+^ GPNMB^+^ TREM2^+^ SPP1^+^ MFs in injured livers. Those MFs colocalize with lipid-rich apoptotic hepatocytes and desmin^+^ areas (=lipid-associated MFs or LAMs, Table 2), and proportionally correlate with the degree of fibrosis [94,95,98,108,194]. A similar MF phenotype has been described in adipose tissue in HFD-fed mice [108] and hepatic fibrotic scars (‘scar-associated MFs’ (=SAMs), Table 2) in carbon tetrachloride-treated mice [64]. Importantly, this phenotype was also described in NAFLD and cirrhosis patients [64,108]. More recently, Daemen et al. showed that the expression levels of most of these LAM/SAM signature markers increase with increasing SPP1 expression, at least in HFD-fed mice [98]. Future studies investigating the functional role of MoMFs that express varying levels of LAM/SAM markers are needed to elucidate their role in fibrogenesis and fibrosis resolution.

Polarization of hepatic MFs could present a more attractive therapeutic approach compared to blocking monocyte recruitment or complete deletion of all phagocytic cells. Numerous preclinical studies have reported on this potential. Saturated fatty acids, prominently present in Western dietary patterns, are known to induce pro-inflammatory gene expression in MFs [195]. Blocking macrophage scavenger receptor 1 (MSR1) in vitro reduced saturated fatty acid-induced inflammatory gene expression in primary MFs and in vivo administration of an anti-MSR1 antibody ameliorated liver inflammation in a NAFLD mouse model [155,180]. Thrombospondin-1 (TSP1) can induce pro-inflammatory gene expression in MFs and MF-specific TSP1^−/−^ mice showed reduced liver injury in experimental NAFLD, suggesting MF-specific TSP1-modulation as a therapeutic strategy [196]. Various other targets have been proposed in NAFLD, of which their in vivo ameliorating effects are reported to be mediated by modulating inflammatory gene expression in MFs, including glucocorticoid-induced leucine zipper [197], hypoxia-inducible factor 1α [198,199,200], vitamin D receptor [201], farnesoid X receptor [202] and peroxisome proliferator-activated receptors (PPARs) [185]. Interestingly, molecules not specifically targeting MFs were also described to influence MF polarization in the NAFLD liver. Resveratrol, protective in experimental ALD, was reported to ameliorate NAFLD in mice fed a HFD via favoring M2 polarization of hepatic MFs. Moreover, in vitro experiments in this study showed that M2-polarized MFs can induce M1 MF apoptosis [203]. Similarly, nobiletin, a natural polymethoxylated flavone, significantly increased the proportion of M2 MFs and the expression of anti-inflammatory factors in vivo and in vitro and moreover ameliorated hepatic damage in MCD-fed mice [204]. Importantly, many of these proposed compounds, molecules or MF-specific drug targets require further preclinical and clinical investigation. Currently, lanifibranor, a pan-PPAR agonist, showed promising preclinical results, including the attenuation of saturated fatty acid-induced inflammatory MF polarization in vitro and ameliorated liver injury in Western diet-fed mice and is in phase 3 (NCT04849728) [185,205].

### 4.2. Alcoholic Liver Disease

Nearly half of all cirrhosis-related mortality is caused by alcohol and worldwide overconsumption of alcohol poses a major economic and societal cost. ALD comprises a pathological spectrum ranging from steatosis, alcoholic hepatitis (AH), to cirrhosis and its complications [2,206,207]. A fatty liver is commonly observed in most drinkers exceeding 40 g of ethanol per day, but only up to one third will develop AH. Around 10% of chronic AH patients will progress to cirrhosis and 1–2% of cirrhotic patients will develop HCC [208]. Although cessation of alcohol intake is by far the most effective treatment to date, abstinence in alcohol-dependent individuals is not straightforward and moreover not always linked to a positive outcome [209]. Corticosteroid therapy was shown to improve survival in severe AH, although only in select patients. For patients with decompensated alcohol-related cirrhosis, liver transplantation, to date, remains the most effective therapy [210]. 

The direct toxic effect of ethanol on hepatocytes has long been recognized (extensively reviewed elsewhere, [211]). Injured hepatocytes release DAMPs which activate hepatic MFs and induce pro-inflammatory cytokine and chemokine secretion, in turn facilitating neutrophil and bone marrow-derived MF (BMDM) recruitment to the liver [212]. Increased MF numbers are found in the livers of patients with acute and chronic ALD, and inflammatory cytokines and monocyte-recruiting chemokines are elevated in the serum of AH or alcohol-related cirrhosis patients. Hepatic gene expression levels and serum levels of MF activation markers correlate with the stage of cirrhosis [125,213]. Moreover, serum levels of MF activation markers, including sCD163 and sCD206, are associated with increased mortality in AH patients [213]. Nonetheless, it is unclear whether the total amount of activation markers only reflects the proportion derived from hepatic MFs or if other immune cells, including intestinal ones, also contribute.

Alcohol, typically consumed via ethanol-containing beverages, initially interacts with the complex environment of the intestinal tract [214]. Alcohol and its metabolites disrupt the mucus layer, are toxic to the intestinal epithelium and have a detrimental effect on TJs [215,216,217,218,219]. Intestinal expression of TJ proteins, including occludin and zona occludens-1, was reported to be reduced in ALD murine models and increased serum levels of gut permeability markers were found in ALD patients [220,221]. Besides barrier disruption, chronic ethanol ingestion also causes gut microbial dysbiosis and it was recently shown that human fecal microbial signatures can discriminate between alcohol consumption and AH, although no correlation was found with ALD severity [222]. While a single alcohol binge did not alter the gut microbiome in healthy young human individuals [223], chronic alcohol intake resulted in bacterial overgrowth and dysbiosis in both alcoholic patients and rodent models [224,225]. In conventional mice, a seven-day alcohol feeding regimen caused microbial dysbiosis and oral transplantation of this enterotype into germ-free mice induced intestinal inflammation, leaky gut, and liver injury [226], thus suggesting a pathogenic role for alcohol-exposed microbiota. In germ-free mice, a single alcohol binge resulted in more liver injury when compared to conventional mice, suggesting a protective role for eubiotic gut microbiota in ethanol-induced liver injury [227]. Further research aiming to identify specific pathogenic and protective gut microbes and their function are warranted. Importantly, given the often conflicting outcomes of ALD studies in rodents, concentration, exposure duration, species, strain, and model used should be carefully considered and specified in future studies [228].

Dysbiosis, bacterial overgrowth, and gut barrier disruption expose LP MFs to an increasing amount of antigens. Interestingly, MF numbers were significantly reduced in the jejunal mucosa of rats and the proximal colon of mice after chronic alcohol intake [229,230]. In contrast, Chen et al. did not observe any changes in the total small intestinal mouse MF pool, although the authors did observe a higher fraction of TNF-α^+^ MFs [225]. Currently, modulation of intestinal microbiota and enhancing gut barrier integrity are being explored as therapeutic strategies in ALD [231]. In this regard, dietary intervention, either via pre- and/or probiotics, specific food intake, nutrient supplementation, or a combined approach [230,232,233,234,235,236,237,238,239,240,241], are the most extensively tested approaches thus far, both in ALD models and patients.

Alcohol-induced intestinal barrier disruption leads to increased translocation of gut microbial products which reach the liver via the portal vein and can activate hepatic MFs. In an acute model of alcohol exposure in mice, 30% of hepatic MFs (F4/80^+^) were depleted after 3 days and restored 7 days later [104]. Mice exposed to prolonged ethanol-feeding (4 weeks) showed an increase in CD11b^hi^ F4/80^int^ Ly6C^+^ cells whereas CD11b^lo^ F4/80^hi^ Ly6C^−^ cells were decreased [242]. Importantly, these Ly6C^+^ cells were further divided into Ly6C^hi^ and Ly6C^lo^ cells with pro-inflammatory and restorative properties, respectively. A recent multi-omics approach in the same animal model revealed that Ly6C^lo^ MoMFs versus their high expressing counterparts were enriched with phagocytosis-related and apoptosis-regulating proteins, and contained higher levels of glycerophospholipid metabolites, which together indicates an anti-inflammatory profile [243]. Interestingly, these shifts in the monocyte-MF pool are reported to be most outspoken in female mice, and the increase in both Ly6C^lo^ and Ly6C^hi^ MoMFs was shown to be dependent on IFN signaling [244]. Blocking monocyte recruitment has been explored as therapeutic strategy in experimental models for ALD. The CCR2/CCR5 dual-inhibitor CVC [245] and TREM1 peptide inhibitors [246] were shown to reduce MF infiltration and ameliorated alcohol-induced steatosis and hepatic injury in mice. However, these receptor-specific strategies remain to be explored in human ALD patients.

As for NAFLD, modulation of MF phenotype in ALD has also been explored as potential therapeutic strategy. Li et al. recently showed that ethanol promotes a pro-inflammatory hepatic MF phenotype via zinc finger SWI2/SNF2 and MuDR-domain containing 3 (ZSWIM3) promotor hypermethylation which resulted in enhanced NF-κB signaling. Restoration of ZSWIM3 levels, mediated by decitabine, a demethylating agent, alleviated ethanol-induced liver injury and reduced the in vivo inflammatory response [247]. Activation of the cannabinoid-2 receptor on hepatic F4/80^+^ MFs promoted an anti-inflammatory phenotype and resulted in autophagy-mediated protection against ethanol-induced liver injury [248]. Two other independent studies using myeloid-specific KO models for *atg5* and *atg7* corroborated the protective role of MF autophagy in alcohol-fed mouse models [249,250]. Alcohol-induced impaired autophagy was shown to result in increased serum extracellular vesicle (EV) levels in vivo and enhanced EV release in hepatocytes and MFs in vitro [251]. The phenotype of hepatic MFs can be modulated via EV-mediated hepatocyte-MF communication. *In vitro* studies have demonstrated that alcohol-damaged hepatocytes can polarize MFs via EVs enriched with miRNA-122 [252], CD40L [253] or HSP90 [254]. Injecting circulating EVs from alcohol-exposed mice into naive counterparts showed hepatic uptake, expansion of the F4/80^hi^ CD11b^int^ MF pool and resulted in an increased expression of TNF-α and IL-12/23 in liver MFs [254]. EVs were even implied in long distance priming of BMDMs and imprinted an EV cargo-dependent phenotype [252,255,256]. In humans, serum levels of miRNA27a-containing EVs are reported to be elevated in ALD patients [255] and the overall serum miRNA signature can discriminate AH patients from healthy controls [257]. However, the therapeutic potential of targeting EVs and their specific cargo in human ALD remains to be explored.

### 4.3. Primary Sclerosing Cholangitis

PSC, a prototype of CLD hallmarked by gut–liver axis disruption, is a rare and progressive liver disease in which multifocal fibro-inflammatory bile duct strictures lead to cholestasis and progressive liver injury. PSC typically affects middle-aged men who are often coincidentally diagnosed given the frequent asymptomatic disease course [258,259,260]. To date, approved pharmacological approaches are lacking and most PSC patients eventually require a liver transplant [261,262]. Up to 80% of PSC patients present with IBD, comprising Crohn’s disease and ulcerative colitis (UC), with the vast majority diagnosed with colitis. In contrast, only up to 10% of IBD patients develop PSC [263,264]. PSC-colitis phenotypically differs from classical UC and it is currently accepted that PSC, UC, and PSC–UC are three distinct disease manifestations [264]. What drives PSC and its concomitant presentation with IBD remains unclear, but aberrant mucosal lymphocyte homing [265], anomalous BA metabolism/signaling [266,267], gut barrier disruption [268,269] and intestinal dysbiosis are proposed in the disease pathogenesis.

In humans, PSC is highly associated with gut microbial dysbiosis and differential gut microbial signatures can discriminate PSC patients from UC patients and healthy subjects. However, the majority of observational gut microbiome studies did not find a PSC–IBD versus PSC distinctive signature [270,271]. Colonic inflammation in PSC–IBD was recently found to be characterized by increased Th17 cell infiltration, similar as in UC. However, the PSC–IBD transcriptome was distinct from UC, with increased dysregulation of multiple BA-regulated pathways, which was potentially mediated by the corresponding differences in gut dysbiosis [272]. Both microbial dysbiosis and intestinal inflammation are associated with intestinal barrier failure and markers for impaired barrier are elevated in PSC patients [268,269]. Where anti-inflammatory drugs including anti-TNF monoclonal antibodies have shown limited potential [273], targeting the gut microbiome in PSC has shown more promising clinical outcomes. Indeed, antibiotics, especially vancomycin, significantly reduce cholestasis markers and lower the Mayo risk score, as published in a recent meta-analysis [274]. Besides antibiotics, fecal microbial transplantation also seems promising as three out of ten PSC patients showed a 50% reduction in alkaline phosphatase levels in an open-label pilot study [275]. Obviously, this approach still requires additional clinical validation and might require more rigorous safety testing.

Several studies in various experimental cholestasis models have described intestinal homeostasis disruption and demonstrated how gut–liver axis dysregulation, including altered gut microbiota, can contribute to liver injury. Intestinal dysbiosis has been described in common bile duct ligation (CBDL) [276,277], 3,5-diethoxycarbonyl-1,4-dihydrocollidine (DDC), α-naphthylisothiocyanate (ANIT) [278] and multidrug resistance-2 KO (Mdr2^−/−^) [279] cholestasis models as well as the NOD.c3c4 biliary inflammation model [280]. Germ-free analogues of these models showed either reduced (ANIT, DDC [278], NOD.c3c4 [280]) or aggravated liver injury (CBDL [281], Mdr2^−/−^ [282]). In contrast to the protective role of gut microbiota in Mdr2^−/−^ reported by Tabibian et al. [282], another study using the same model showed that antibiotic treatment ameliorates liver injury and fecal microbial transplantation of the Mdr2^−/−^ enterotype into wild-type mice spontaneously induced liver injury, underscoring a pathogenic role for gut microbiota in cholestatic liver injury [279]. Nakamoto et al. identified a triad of bacterial strains, including *Klebsiella pneumonia*, in human PSC fecal cocktails which increased gut leakiness via pore formation and resulted in liver inflammation, underscoring the potential of PSC microflora to negatively impact liver homeostasis via barrier disruption [283]. Barrier disruption can also aggravate fibrosis, as shown in Mdr2^−/−^ [284], but not in heterozygous Mdr2^+/−^ mice [285]. Inversely, established cholestatic liver disease enhanced susceptibility to dextran sodium sulfate (DSS)-induced colitis, which was attenuated by ursodeoxycholic acid administration. However, the authors did not investigate the mechanisms associated with the observed aggravated phenotype and the reciprocal impact of gut–liver disease needs to be further explored [286].

While intestinal MFs have been extensively characterized in colitis mouse models and human colonic biopsies, and were demonstrated to play crucial roles in gut inflammation and barrier preservation [51,142,287] (reviewed in [13]), research on the role of colonic MFs in cholestatic liver-injury remain scarce. So far, increased infiltration of F4/80^+^ cells has been reported in the ileum and colon of cholestatic *Mdr2*^−/−^ mice [279] and general MF depletion improved intestinal barrier function and simultaneously ameliorated liver injury in the ANIT model [278]. Importantly, to date, no single liver or combined gut–liver insult model can fully capture human PSC pathophysiology [288,289], but such a model is highly anticipated.

Though PSC is described as an immune-mediated liver disease, research on the role of PSC-associated hepatic MFs has only recently accelerated. CD68^+^ MFs accumulate in perisinusoidal and periportal regions in the liver of PSC patients irrespective of their IBD status and CD68^+^ CD206^+^ MFs are more abundantly present in the livers of PSC patients compared to other liver diseases [290,291]. Hepatic CD206 expression further correlates with hepatic injury [85,292] and circulating sCD206 levels can predict transplant-free survival in PSC patients [293].

Accumulation of hepatic F4/80^+^ or CD68^+^ cells has also been shown in various acute and chronic cholestasis rodent models [85,294,295,296]. Cholestatic livers of BV6 and Mdr2^−/−^ mice showed an increased proportion of F4/80^lo/+^ CD11b^hi^ MFs [85,279], while F4/80^+^ CD11b^int^ MFs remained stable [85]. In CBDL mice, Ly6C^lo^ CD11b^hi^ MFs were also enriched, while Ly6C^lo^ CD11b^int^ MFs were depleted [297]. Studies have demonstrated that reactive cholangiocytes can coordinate inflammation and tissue repair via monocyte/MF recruitment. In turn, MFs can modulate cholangiocyte proliferation and thus perpetuate cholangiopathy-associated liver injury [295,298,299,300]. This underscores the potential of targeting monocyte/MF recruitment in PSC. Indeed, genetic or pharmacological targeting of the CCL2/CCR2 axis has resulted in improved disease outcomes in PSC models [85,96]. In human PSC however, only a modest reduction in alkaline phosphatase levels was observed after 24 weeks of CVC treatment in the PERSEUS trial. In contrast to the CENTAUR (NASH) study where CVC was most efficacious in advanced fibrosis, only mild-to-moderate fibrotic PSC patients were included [301]. Of note, CVC treatment potentiated the ameliorating effect of all-trans retinoic acid, a BA pool-modulating agent, in murine experimental cholestasis [302], but this approach has not been tested in humans.

Several studies using liposome-encapsulated clodronate or gadolinium chloride as a MF depletion strategy have shown conflicting outcomes in murine cholestasis and biliary injury models and therefore our understanding of the relative pathological contribution of each liver MF subtype remains elusive [295,297,300,303,304,305,306]. In order to more accurately pinpoint which MF populations initiate, perpetuate, or resolve cholestatic liver injury in PSC, the application of high resolution techniques using the latest subset-specific markers, in conjunction with functional studies and subset-specific conditional depletion are warranted.

Preclinical and in vitro evidence suggests that modulating the polarization of MFs could also have translational potential in PSC. Isolated cholangiocytes from Mdr2^−/−^ mouse livers were shown to induce a pro-inflammatory phenotype in BMDMs via DAMP-containing EVs [307]. Similarly, cholangiocyte cell line-derived EVs containing non-coding RNA HN19 induced a pro-inflammatory phenotype in isolated hepatic MFs and BMDMs. Interestingly, H19 KO in Mdr2^−/−^ and CBDL-operated mice alleviated hepatic injury characterized by reduced F4/80^+^CD11b^hi^ MoMF influx [308]. H19′s role in CLD is so far mostly explored in HCC, where it is implicated in tumor initiation, promotion, and suppression [309]. Far fewer studies have explored its role in various CLD etiologies, including PSC, and could represent a promising avenue of research. Wntless is a chaperone required for Wnt secretion and a dysregulated Wnt/b-catenin pathway is implicated in CLD [310]. In the DDC model, myeloid-specific KO of Wntless aggravated liver injury and resulted in hepatic F4/80^+^ MF accumulation. Additionally, BMDMs or isolated liver MFs from Wntless KO DDC mice were found to be more pro-inflammatory compared to their wild-type counterparts [284,311]. Myeloid-specific CD39 KO also aggravated liver injury in the DDC model, but the authors did not find a difference in M1/M2 gene expression and suggest that an intermediate MF subtype contributes to disease amelioration [294]. Further studies should explore the therapeutic potential of targeting these genes and/or their gene products in specific liver MF subsets in experimental PSC.

### 4.4. Liver Fibrosis/Cirrhosis

The progression of NAFLD, ALD, and PSC is associated with fibrogenesis, which can lead to cirrhosis [8,312]. The disturbed hepatic architecture and vasculature in cirrhosis is characterized by the presence of regenerative nodules and extensive fibrous scar tissue formation, and associated with impaired liver function, portal hypertension, and an increased risk for HCC development [8,313,314]. Transition of compensated cirrhosis to a decompensated state occurs in 4–12% of cirrhosis patients yearly, and is associated with the occurrence of at least one of the following life-threatening complications: variceal hemorrhage, ascites, jaundice, hepatic encephalopathy, and hepatorenal and hepatopulmonary syndrome [2,313,314,315]. The 5-year mortality rate without liver transplantation following decompensation is 85% [2,313]. In addition to the effect on multiple other organ systems, liver cirrhosis has been associated with gastrointestinal dysfunction, underscoring the contribution of the gut–liver axis in cirrhosis development and deterioration [316].

The gut microbiome of liver cirrhosis patients displays an etiology-specific imbalance between beneficial and potentially pathogenic bacteria and small intestinal bacterial overgrowth. Together with increased intestinal permeability, intestinal inflammation, and immune disturbances, this increased intestinal burden of potentially harmful bacteria results in both quantitative and qualitative alterations of intestinal BT, which contributes to the progression of cirrhosis by inducing chronic activation of the hepatic and systemic immune system [317,318]. The interaction of translocated MAMPs and TLRs, expressed on the surface of hepatic MFs, results in the production of several inflammatory cytokines, chemokines, nitric oxide, and reactive oxygen species. This further contributes to liver inflammation through the recruitment of immune cells, including CD4^+^ T cells and monocytes, increased vascular permeability, and the induction of hepatocyte apoptosis and necrosis. The resulting disturbance of the hepatic architecture and vasculature eventually leads to portal hypertension, which causes intestinal hypoperfusion and subsequent hypoxia-mediated intestinal damage, and thereby also promotes BT. Consequently, fibrosis-mediated liver damage is not only a result of, but also contributes to gut barrier disruption-mediated BT [319,320]. In compensated liver cirrhosis, BT is limited to MAMPs, whereas during decompensated cirrhosis, intestinal permeability has increased to the extent of viable BT [22]. Activated intestinal MFs play a significant role in intestinal barrier disruption and subsequent pathological BT in liver cirrhosis, through the secretion of cytokines and chemokines responsible for monocyte recruitment, activation, and polarization. Indeed, in human cirrhosis, intestinal CD14^+^ TREM1^+^ iNOS^+^ MFs secrete several chemokines, including IL-8 and CCL2, resulting in the recruitment of circulating monocytes in the LP and release of IL-6 and nitric oxide, which induce the expression of the gut barrier disruption-associated TJ protein claudin-2. The subsequent increase in intestinal permeability opens the gateway for intestinal products to enter the LP, bypass the vascular endothelium of the portal vein and eventually reach the liver [145,321].

In the liver, in addition to exposure to persistent hepatic injury-associated DAMPs, increased intestinal translocation of harmful MAMPs can activate resident KCs through pattern recognition receptors [134,175,317,319], resulting in CXCL1-, CXCL2- and CXCL8-dependent recruitment of neutrophils, as well as the recruitment of Ly6C^hi^ monocytes, mediated by several CCLs, including CCL2, CCL3, and CCL5 [12,126,322,323]. The crucial role of monocyte recruitment in fibrogenesis has been emphasized by several studies in which pharmacological inhibition or genetic deletion of certain CCLs/CCRs in experimental mouse models of fibrosis resulted in attenuation of disease [139,324,325]. In mice, both resident KCs and infiltrated MoMFs promote transdifferentiation of HSCs into extracellular matrix-producing myofibroblast-like cells through secretion of several profibrotic factors, including TGF-β, PDGF, galectin-3, and SPP1, thereby driving the progression of liver fibrosis [94,139,140,322]. In addition, murine hepatic MFs are also implicated in the survival of these activated HSCs through secretion of the inflammatory cytokines IL-1β and TNF-α [326]. In humans, pro-inflammatory monocyte-derived CD14^+^ CD16^++^ MFs are observed during fibrosis progression, and have been shown to activate HSCs in vitro [327]. These MFs are also recruited in a CCR2-dependent manner and are derived from either classical CD14^++^ CD16^−^ or non-classical CD14^+^ CD16^++^ circulating monocytes [38,328]. *In vitro* studies suggest that local differentiation of the infiltrated monocytes is driven by high, chronic liver inflammation-associated levels of TGF-β and IL-10 [328]. In addition, granulocyte-macrophage colony-stimulating factor (GM-CSF) promotes monocyte differentiation towards fibrosis-associated CD206^+^ MoMFs [329]. Besides pro-inflammatory cytokines, intrahepatic CD14^+^ CD16^++^ MFs also secrete the profibrotic cytokine IL-13 and several chemokines [328]. These secreted factors may contribute to fibrogenesis through promotion of HSC activation and survival (TNF-α), myofibroblast proliferation (IL-6), myofibroblast-mediated collagen production (IL-6 and IL-13) and monocyte recruitment (IL-8, CCL2, CCL3 and CCL5) [328,330]. Thus, despite the inability of murine liver fibrosis models to fully mimic the major pathological and molecular features of human disease, some important parallels exist between both species, including the involvement of TREM1 in KC activation and the role of the CCL2–CCR2 axis in monocyte recruitment [105,322,331]. With regard to the latter, as described earlier, the dual CCR2/CCR5 antagonist CVC showed promising results in both experimental animal models and NASH patients with severe fibrosis [86,332,333,334].

Despite their fibrosis-promoting role, hepatic MFs also play a critical role in fibrosis resolution. Indeed, due to their heterogeneity and plasticity, liver MFs are able to exert context- and time-dependent fibrolytic functions as well [322]. In mice, restorative MFs arise from the post-phagocytic phenotype switch of Ly6C^hi^ towards Ly6C^lo^ hepatic MFs [37,67,140,330,335]. They orchestrate the resolution of fibrosis mainly through the expression of extracellular matrix-degrading matrix metalloproteinases (MMPs), including MMP9, MMP12, and MMP13. In addition, these restorative MFs contribute to fibrolysis through increased expression of several phagocytosis-related genes, elevation of local growth factor levels, including insulin-like growth factor-1, VEGF, and M-CSF, and promotion of myofibroblast apoptosis by means of TRAIL expression [37,330,335,336]. Confirming the presence of functionally distinct hepatic MF subpopulations in fibrotic liver tissue, Duffield et al. demonstrated that MF depletion in experimental liver fibrosis has an antifibrotic effect during fibrosis progression and a profibrotic effect during fibrosis resolution [337]. Critical involvement of MoMFs in fibrosis resolution was also demonstrated by Mitchell et al., who showed that CCR2 deficiency not only hinders fibrogenesis, but also fibrolysis [338]. In addition, Karlmark et al. showed that the interaction of CX3CR1, expressed on circulating monocytes and hepatic MFs, and CX3CL1, primarily derived from hepatocytes and HSCs, limits hepatic inflammation and fibrogenesis in mouse models of liver fibrosis, and this by regulating the differentiation and survival of MoMFs. Importantly, hepatic expression of both CX3CR1 and CX3CL1 is downregulated in cirrhotic patients [339].

During advanced liver disease, profibrotic and restorative MFs, together also referred to as SAMs (Table 2), seem to co-exist in the liver. The progression of fibrosis/cirrhosis is therefore dependent on the balance between these two MF populations [103,330,340,341]. As liver fibrosis, and even cirrhosis, have been shown to be reversible processes, modulation of the composition of the hepatic MF pool in favor of fibrosis resolution seems to be a reasonable therapeutic strategy [342]. Interestingly, Ma et al. demonstrated the therapeutic potential of shifting the balance between these MF populations towards fibrosis resolution by means of cytotherapy with M1-polarized BMDMs in experimental liver fibrosis [343]. The fibrosis resolution was primarily due to antifibrotic modulation of the microenvironment including recruitment of restorative Ly6C^lo^ MFs and HSC apoptosis-mediating NK cells. In this view, despite the feasibility of ex vivo differentiation of cirrhotic patients’ CD14^+^ monocytes into pro-resolution MFs, pharmacological modulation of MF recruitment or polarization seems to be a more straightforward option [10,323,343]. A broad range of hepatic MF-targeting strategies have shown therapeutic potential in experimental animal models of liver fibrosis, but only a handful of them have already advanced to clinical trials [11,134,140]. Further efforts to improve our understanding of MF heterogeneity, plasticity, and function during liver fibrosis in order to promote the identification of novel, promising treatment options are thus of need. In light of this, Ramachandran et al. recently uncovered a conserved scar-associated pro-fibrogenic TREM2^+^ CD9^+^ MF subpopulation, originating from circulating monocytes, in human and murine fibrotic liver tissue and further studies will now have to reveal the function of these cells [64].

### 4.5. Hepatocellular Carcinoma

HCC is the most common form of primary liver cancer, the 6th leading cancer type worldwide and the 4th most common cause of global cancer-related mortality [344]. In terms of risk factors for the development of HCC, chronic HBV and HCV infections are by far the most common ones, followed by excessive alcohol use, obesity, diabetes mellitus and aflatoxin exposure [2,344,345,346]. In 80–90% of HCC patients, hepatocarcinogenesis occurs in the setting of background cirrhosis [345,347]. Indeed, chronic inflammation acts as a favorable preneoplastic setting and HCC development marks the end of the CLD spectrum [348,349]. Cirrhosis is associated with necroinflammation and chronic hypoxia-related oxidative stress, inducing repetitive injury-mediated regeneration of hepatocytes and the subsequent appearance of regenerative nodules. The genomic instability characteristic of the precancerous lesions eventually initiates the development of HCC [345,348,350]. The poor prognosis, rapid disease progression and limited effectiveness of available systemic treatment options emphasize the importance of high-quality research in the search for promising targets for HCC therapy [344].

In CLD-associated HCC, the combination of intestinal dysbiosis and gut barrier disruption results in prominent pathogenic BT, which promotes HCC by mediating hepatic inflammation and fibrogenesis [351,352,353]. Indeed, penicillin-induced enteric dysbacteriosis and DSS-induced intestinal inflammation promote tumor formation in experimental HCC [354,355]. Moreover, TLR4-mediated LPS-induced activation of KCs contributes to HCC progression through the expression of the hepatomitogen epiregulin, which reduces hepatocyte apoptosis [352,356,357]. However, the relative contribution of KCs in hepatic TLR4 signaling in context of HCC is still under debate, as this signaling function could also be primarily attributed to HSCs and hepatocytes [352,356,358]. As is the case for fibrogenesis, TREM1 also appears to play an important role in hepatocarcinogenesis-mediated KC activation [359]. In addition to LPS, the gram-positive microbial component lipoteichoic acid and the secondary BA deoxycholic acidare are involved in HCC promotion as well. Increased translocation of these components to the liver leads to the production of pro-inflammatory cytokines and the creation of a tumor-promoting microenvironment [335,351,352,357]. In addition to its involvement in HCC promotion, the gut–liver axis is also implicated in HCC progression, as IL-25 secretion by gut microbiota dysbiosis-induced hyperplastic colonic epithelial tuft cells induces M2-like polarization and CXCL10 expression of hepatic MFs in the tumor microenvironment [360].

HCC is a typical inflammation-related cancer, characterized by continuous cytokine expression and hepatic infiltration of immune cells [361]. Tumor-associated MFs (TAMs, Table 2) are a crucial part of the tumor microenvironment and have been correlated with poor prognosis [11,71]. However, which cells contribute to the TAM pool has only partly been unraveled. We and others showed that HCC is characterized by KC depletion and a simultaneous rise of MoMFs, at least in mice [60,362,363]. However, we also showed that, in addition to the widely accepted contribution of infiltrating MoMFs to the TAM pool in HCC, KCs also express markers that are typically associated with TAMs [128]. The group of Ye et al. recently showed that inhibition of monocyte-to-TAM differentiation leads to expansion of a pro-tumorigenic KC-like TAM population [364]. It remains however to be confirmed whether these findings in murine HCC models can be translated to human disease [128]. Zhang et al. recently performed scRNA-seq on HCC patient-derived liver tissue, describing two distinct tumor-enriched MF populations, namely THBS^+^ myeloid-derived suppressor cell (MDSC)-like and C1QA^+^ TAM-like MFs (Table 2). Further studies focusing on elucidating specific TAM populations and their functions should ensue [119].

Tumor microenvironmental signals are crucial determining factors for the polarization and functionality of TAMs. In HCC, TAM recruitment and activation depends, among other things, on tumor cell expression of glypican-3 and several chemokines, including CCL2 and M-CSF [140,361,364]. TAMs contribute to tumor growth, angiogenesis, and metastasis through the production of cytokines, including IL-1β, IL-6, IL-10, IL-17, and TNF-α, chemokines, like CCL2, CCL17, CCL22, CXCL10, and CXCL12, growth factors, such as PDGF-β, VEGF, TGF-β, and other factors, including MMPs, SPP1, and cyclooxygenase 2 [11,140,189,361]. Moreover, TAMs are able to induce neoplastic transformation by causing DNA damage and gene mutations through the release of free radicals, such as reactive oxygen species and nitric oxide [361]. Another mechanisms by which hepatic MFs regulate HCC promotion, is through induction of the expression of negative costimulatory immune checkpoints, including programmed death protein 1 (PD-1) and PD-L1 [365,366,367,368]. Through secretion of IL-1β and IL-10 in the peritumoral stroma, they upregulate the expression of PD-L1 on their own surface, as well as on HCC cells. Subsequent interaction with PD-1, expressed on the surface CD8^+^ T cells, leads to suppression of antitumor T cell immunity [367,368]. In addition, hepatic TREM1^+^ TAMs inhibit cytotoxic T cells through the attraction of Tregs [369,370]. In contrast, hepatic MFs have also been shown to participate in the antitumor response in a mouse model of oncogene-induced hepatocyte senescence. In this setting, following CCL2 secretion, pre-malignant hepatocytes were cleared by recruited CCR2^+^ pro-inflammatory MoMFs, thereby preventing HCC initiation [371,372]. However, in established HCC, peritumoral CCR2^+^ MoMFs enhance tumor growth through the inhibition of NK cells [371]. Despite the fact that CCL2 overexpression has been correlated with poor HCC patient prognosis, and promising results of CCR2 KO/antagonism in experimental HCC, translation of these encouraging findings to the clinic should be done with certain caution, as the CCR2/CCL2 axis thus seems to play a context-dependent tumor-promoting or protective role in hepatocarcinogenesis-mediated monocyte recruitment [109,373].

TAMs have often been labeled as immunosuppressive, tumor-promoting M2-like polarized MFs, but the HCC-associated TAM pool is actually very heterogeneous, as multiple phenotypes exist, depending on the disease stage and the intratumoral location [71,110,361]. Despite being an overly simplistic representation of the actual situation, due to the limited knowledge on HCC-associated TAM heterogeneity, TAMs are still commonly classified as either M1-like or M2-like MFs [374]. In human HCC, pro-inflammatory M1-like CD68^+^HLA-DR^+^ TAMs are more abundant in the early stages of HCC progression and undergo a phenotypic switch towards immunosuppressive M2-like CD68^+^ HLA-DR^−^ TAMs once the tumor is more established [71,361,374]. Furthermore, MFs with a more pro-inflammatory CCR2^+^ phenotype are rather localized perivascularly and in the peritumoral stroma, contributing to neoangiogenesis and the pro-inflammatory tumor microenvironment, while M2-like CCR2^−^ TAMs are rather located in hypoxic areas and cancer nests [110,361,374]. M2-like-polarized TAMs, characterized by the expression of CD206 and CD163, contribute to HCC progression through tumor immune evasion and are associated with a poor prognosis [110,361,374,375,376]. They display high levels of IL-10 and arginase, low levels of pro-inflammatory cytokines and free radicals, and low antigen-presenting abilities [361]. M2 polarization is, among other mechanisms, promoted by increased TGF-β levels in the tumor microenvironment and the subsequent enhancement of TIM-3 expression on the surface of TAMs [377]. M1-like TAMs, characterized by the expression of CD86, CD11c, and TLR4, show functional diversity in HCC, as they play a tumoricidal role by exerting anti-tumor cytotoxic activity but also promote tumorigenesis in the early stages of HCC through the release of several pro-inflammatory cytokines, which contributes to an inflammatory tumor microenvironment [71,375,378]. Moreover, it has been demonstrated that M1-like-polarized TAMs can contribute to suppression of T cell-mediated antitumor immunity and are able to promote angiogenesis and metastasis in HCC as well [71,368]. Of note, recently, a CD38^+^ TAM subpopulation with a pro-inflammatory M1-like phenotype was described to be positively associated with improved HCC prognosis [379]. Based on the above findings, pharmacological targeting of M2 polarization appears to be an attractive MF-based therapeutic approach for HCC, and several preclinical studies evaluating different TAM re-educating agents have indeed confirmed the therapeutic potential of this strategy [380]. However, as the full complexity of TAM heterogeneity and plasticity in HCC pathogenesis is still not completely understood, it will be crucial to gain further insights into MF biology and function in HCC, and CLD in general, for successful translation of these promising preclinical results to the clinic [189].

## 5. Conclusions

Both intestinal and hepatic signals contribute to CLD-associated inflammation, fibrogenesis, and hepatocarcinogenesis. While the role of hepatic MFs in liver disease is extensively being explored, little is known about the role of intestinal MFs as contributors to liver disease. In addition, the involvement of gut barrier disruption, specific gut-derived signals and associated intestinal dysbiosis in hepatic MF recruitment, polarization, and function is understudied, but may represent a plethora of attractive (immuno)therapeutic targets, especially since CLD is highly associated with gut–liver axis disruption. Particularly, whether restoration of qualitative and quantitative dysbiotic alterations of the gut microbiome can be translated into effective CLD treatment requires further investigation. Despite the multitude of promising preclinical findings regarding the therapeutic potential of liver MFs as targets, some crucial hurdles remain to be overcome for successful translation to the clinic. While inhibition of MF recruitment might not be a good therapeutic strategy, modulation of MF polarization might be a more eligible therapeutic approach. To that end, further investigations of hepatic MF heterogeneity, plasticity, and context-dependent functionality is an absolute need. To date, with regard to CLD, MF heterogeneity is most extensively described in experimental NAFLD and future efforts should extend to other CLD models and human patients. In summary, an increased understanding of the origin, expression profile, function, and interspecies translatability of specific gut and liver MF subpopulations may promote their potential as immunotherapeutic targets in CLD and accelerate the development of MF-targeted drugs.

## Figures and Tables

**Figure 1 cells-10-02959-f001:**
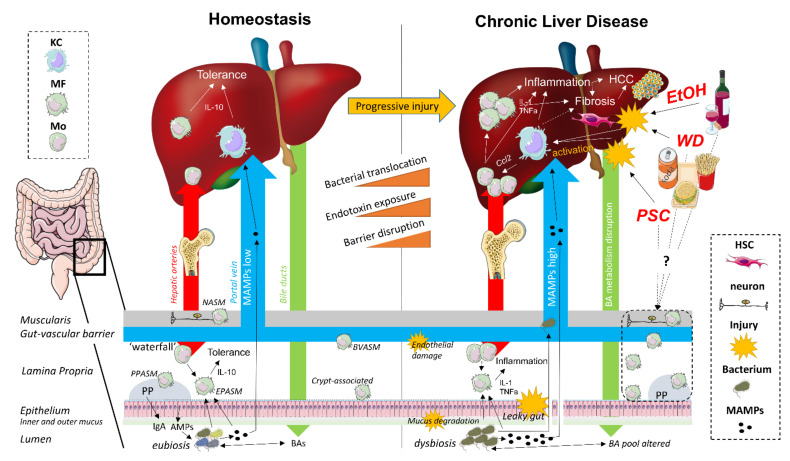
Gut–liver axis in homeostasis and chronic liver disease. Left: In homeostasis, gut microbiota are in a eubiotic state, balanced by IgA and anti-microbial peptides (AMP), and are pivotal in maintaining bile acid (BA) homeostasis. Epithelial-associated macrophage (MF) subsets in the lamina propria are continuously replenished by bone marrow-derived monocytes, which are rendered tolerogenic by eubiosis and physiological levels of microbe-associated molecular patterns (MAMP). The intestinal MF pool further consists of subsets that occupy specific niches. Under homeostatic conditions, the MF pool in the liver mainly consists of Kupffer cells (KC), while small numbers of monocyte-derived MFs (MoMF) and bone marrow-derived monocytes (Mo) are also present. Normal gut–liver axis-associated physiology maintains hepatic immunological tolerance. Right: Progressive injury from different etiologies including alcohol abuse (EtOH), Western diet (WD) consumption and cholestasis (PSC) is associated with dysbiosis, BA pool alterations and intestinal barrier disruption, leading to bacterial translocation and endotoxin exposure. How the intestinal MF pool is altered during CLD progression requires further attention. In the injured liver, Mos and MoMFs infiltrate and induce a shift in the composition of the MF pool. Hepatic MFs are instrumental in CLD-associated inflammation, progressive fibrosis and HCC development. AMPs: anti-microbial peptides; BAs: bile acids; BVASM: blood vessel-associated macrophage; EPASM: epi-thelium-associated macrophage; EtOH: ethanol; HCC: hepatocellular carcinoma; KC: Kupffer cell; MF: macrophage; Mo: monocyte; NASM: neuron-associated macrophage; PP: Peyer’s patch; PPASM: Peyer’s patch-associated macrophage; PSC: primary sclerosing cholangitis; WD: Western diet.

**Table 2 cells-10-02959-t002:** Hepatic macrophage subsets in mouse and human.

Macrophage Subset	Markers	Reference(s)
**MOUSE**		
Resident Kupffer cell (KC)	F4/80^hi^ CD11b^int^ Ly6C^lo^ CX3CR1^lo^ Clec4F^+^ TIM4^+^ Clec2^hi^ VSIG4^+^ CD207^+^ CD163^+^ MARCO^+^	[54,57,58,59,60,64,66,67,68,69,70,71,72,73,74,75,76,77,78,79,80,81,82,83,84,85,86,87,88,89,90,91,92,93,94,95,96,97,98,99,100,101]
*M1-like KC*	CD11c^+^	[54]
*M2-like KC*	CD206^+^ CD317^+^ CD1d^+^	[54]
*Metabolic KC*	CD206^hi^ ESAM^+^	[56]
*AILI-associated restorative KC*	F4/80^+^ CD11b^+^ MHCII^hi^ Ly6C^lo^ MerTK^+^	[88]
Infiltrating monocyte-derived macrophage (MoMF)	F4/80^int^ CD11b^hi^ CX3CR1^hi^ CCR2^+^	[66,67,69,70,71,72,73,74,76,77,79,80,81,82,83,84,85,86,88,90,91,95,96,98,99,100,102]
*Proinflammatory/profibrotic MoMF*	Ly6C^hi^	[67,71,79,80,83,84,85,86,95,102,103,104,105]
*Anti-inflammatory/restorative MoMF*	Ly6C^lo^	[67,71,79,80,83,84,86,95,102,103]
Transitioning macrophage	CX3CR1^hi^ CCR2^hi^ CD11c^+^ Clec4F^−^ TIM4^−^ Clec2^+^ VSIG4^−^	[94,97,98]
*Transitioning LAM*	TREM2^hi^ SPP1^+^ CD9^+^ CD63^+^ GPNMB^+^	[98]
Monocyte-derived Kupffer cell (MoKC)	F4/80^hi^ CD11b^lo/int^ CX3CR1^lo^ CCR2^lo^ Clec4F^+^ TIM4^−^ Clec2^hi^ VSIG4^+^	[57,59,78,87,89,94,95,97,98]
NASH-associated macrophage (NAM)	F4/80^hi^ Clec4F^hi^ TREM2^hi^ CD9^hi^ GPNMB^hi^	[106,107]
Lipid-associated macrophage (LAM)	CX3CR1^lo^ CCR2^lo^ TREM2^hi^ CD9^hi^ CD63^hi^ SPP1^hi^ GPNMB^hi^	[90,95]
ALD-associated M2-like macrophage	F4/80^+^ CD163^+^ CD206^+^	[94,98,108]
Scar-associated macrophage (SAM)	F4/80^+^ CD11b^+^ TIM4^−^ CD9^+^ TREM2^+^	[64]
M1-like tumor-associated macrophage (TAM)	F4/80^+^ CD11b^+^ MHCII^hi^ CD206^−^	[109,110,111]
M2-like tumor-associated macrophage (TAM)	F4/80^+^ CD11b^+^ MHCII^lo^ CD206^+^	[109,110,111]
Liver capsular macrophage	F4/80^+^ CD11b^lo^ CD11c^lo^ MHCII^hi^ CD64^+^ CX3CR1^hi^ Ly6C^lo^ TIM4^−^	[61]
Peritoneal macrophage	CD11b^+^ F4/80^hi^ GATA6^+^ Ly6C^−^ MHCII^−^ CD206^+^ CD64^+^ CD68^+^ CD11c^+^	[62]
**HUMAN**		
Kupffer cell (KC)	CD68^+^ CD11b^−^ CD14^int^ CD32^hi^ CD163^+^ VSIG4^+^ MARCO^+^ TIMD4^+^ CCR2^−^	[53,63,64,79,85,86,112,113,114]
*Tolerogenic KC*	CD163^+^ VSIG4^+^ LILRB5^+^ CD5L^+^ MARCO^+^ HMOX1^hi^	[53,63]
*Inflammatory KC*	CD163^+^ VSIG4^+^ CD1C^+^ FCER1A^+^	[63]
Infiltrating monocyte-derived macrophage (MoMF)	CD68^+^ CD11b^+^ CD14^hi^ CD32^int^ CCR2^+^ S100A9^+^ MAC387^+^ MARCO^−^	[53,79,85,86,112,113,114,115,116]
*HCV-associated pro-inflammatory macrophage*	CD14^+^ HLA-DR^hi^ CD206^+^	[117]
Monocyte-derived Kupffer cell (MoKC)	CD163^+^ MARCO^+^ TIMD4^−^	[64]
ALF-associated restorative macrophage	MerTK^+^ HLA-DR^hi^ CD163^hi^ Tie-2^hi^	[79,88]
Fibrosis-associated macrophage	CD68^+^ CD163^+^ MMP9^+^	[116,118]
Scar-associated macrophage (SAM)	TREM2^+^ CD9^+^ MNDA^+^	[64]
Tumor-associated MDSC-like macrophage	THBS1^+^	[119]
Tumor-associated macrophage (TAM)	CD68^+^ CD14^+^ C1QA^+^	[119,120]
*M1-like TAM*	CD86^+^ CD169^+^ HLA-DR^+^ CCR2^+^	[71,110,121,122,123,124]
*M2-like TAM*	CD163^+^ CD204^+^ CD206^+^ HLA-DR^−^ CCR2^−^ Arg1^+^ MR^+^	[71,110,121,122,123,124]

## Data Availability

Not applicable.

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
