# Peer review of "The Gut–Liver Axis in Chronic Liver Disease: A Macrophage Perspective"

_cells, 2021, doi:10.3390/cells10112959_

Round 1

Reviewer 1 Report

Congratulations on a very well written, comprehensive review of the role of macrophages in chronic liver disease.

A very minor criticism is that it is very long and quite wordy but that is really just the flip side of how comprehensive it is

Author Response

We sincerely thank the reviewer for this acknowledgement. As we aimed to cover different CLD etiologies and progression states, gut-liver axis disruption on various levels, as well as macrophage heterogeneity in both the gut and the liver, our discussion indeed resulted in an extensive manuscript.

Reviewer 2 Report

Dr. Muynck et al in this article review the role of intestinal and hepatic macrophages in chronic liver disease. The article is comprehensive and very well written which this reviewer thoroughly enjoyed reading.

The authors have taken three examples of chronic liver disease namely NAFLD, alcohol-associated liver disease (ALD), and primary sclerosing cholangitis (PSC) to describe the role of gut-liver axis (permeability, microbiome, and bile acids). In addition, the article talks about at length the role of innate immunity especially on intestinal and hepatic macrophages in each of these diseases.  

The authors provide a figure on the general perspective of the role of gut-liver axis in chronic liver disease (Figure 1) and detail on intestinal / hepatic macrophages in humans and mice (Tables 1 and 2). As the pictorial representation is better for readers especially clinicians and translational researchers, if the authors feel feasible, this reviewer suggests the authors to consider providing separate similar figures in each of the three chronic liver diseases, NAFLD, ALD, and PSC. Further, some of the important references on the clinical presentation and burden of the disease are worth mentioning and citing (PMID 34255003, 32022875, and 32246941).  

Author Response

We sincerely thank the reviewer for his comments and suggestions.     
Regarding the figure, we aimed to visually summarize shared etiology-independent gut-liver axis alterations on the intestinal and hepatic level, with specific focus on macrophages. Given the substantial overlap regarding gut-liver axis disruption-associated processes between our selected etiologies, we propose not to split the image in 3 separate ones for publication unless the reviewer feels necessary.   
Concerning the suggested references to include in our revised manuscript, we have added PMID 34255003 (1) on line 409-412 in the introductory paragraph of 4.2 ‘Alcoholic liver disease’. We also included PMID 34255003 (2) on line 269 in the introductory paragraph of 4.1 ‘Non-alcoholic fatty liver disease’. Regarding PMID 32246941 (3), as this proposed manuscript covers oesophageal cancer, we believe this topic is not within the scope of our review.